# Targeted Therapy of Hepatocellular Carcinoma Using Gemcitabine-Incorporated GPC3 Aptamer

**DOI:** 10.3390/pharmaceutics12100985

**Published:** 2020-10-18

**Authors:** Jun Young Park, Ju Ri Chae, Ye Lim Cho, Youndong Kim, Dasom Lee, Jeong Kyun Lee, Won Jun Kang

**Affiliations:** 1Department of Nuclear Medicine, Severance Hospital, Yonsei University College of Medicine, 50-1 Yonsei-ro, Seodaemun-gu, Seoul 03722, Korea; abies60@yuhs.ac (J.Y.P.); JJURI1230@yuhs.ac (J.R.C.); ETOMMI@yuhs.ac (Y.L.C.); 2Aptamer Sciences Inc., 172, Dolma-ro, Bundang-gu Seongnam-si, Gyeonggi-do 13605, Korea; youndong@aptsci.com (Y.K.); dasom@aptsci.com (D.L.); jklee@aptsci.com (J.K.L.)

**Keywords:** hepatocellular carcinoma, aptamer, GPC3, aptamer-drug conjugate, gemcitabine

## Abstract

Hepatocellular carcinoma (HCC) is the most common malignancy of the liver, which can progress rapidly and has a poor prognosis. Glypican-3 (GPC3) has been proposed to be an important diagnostic biomarker and therapeutic target for HCC. Aptamers have emerged as promising drug delivery vehicles because of their high binding affinity for target molecules. Herein, we developed G12msi, a gemcitabine-incorporated DNA aptamer, targeting GPC3, and evaluated its binding specificity and anti-tumor efficacy in GPC3-overexpressing HCC cell lines and murine xenograft models. GPC3-targeted aptamers were selected by using the SELEX process and the chemotherapy drug gemcitabine was internally incorporated into the aptamer. To determine the binding affinity and internalization of the G12msi, flow cytometry and confocal microscopy were performed on GPC3-positive HepG2, Hep3B, and Huh7 cells, as well as a GPC3-negative A431 cell. The anti-tumor activities of G12msi were evaluated with in vitro and in vivo models. We found that G12msi binds to GPC3-overexpressing HCC tumor cells with high specificity and is effectively internalized. Moreover, G12msi treatment inhibited the cell proliferation of GPC3-positive HCC cell lines with minimal cytotoxicity in control A431 cells. In vivo systemic administration of G12msi significantly inhibited tumor growth of HCC HepG2 cells in xenograft models without causing toxicity. These results suggest that gemcitabine-incorporated GPC3 aptamer-based drug delivery may be a promising strategy for the treatment of HCC.

## 1. Introduction

Hepatocellular carcinoma (HCC) is the most common malignancy of the liver and is one of the major causes of cancer-related death globally [1]. HCC mortality is high because most patients are diagnosed at an advanced stage and the overall prognosis of HCC is poor [2]. Treatment strategies for HCC can be broadly divided into two categories: curative (surgical resection, percutaneous ablation, and liver transplantation) and palliative (transcatheter arterial chemoembolization, radioembolization, and systemic therapy) [3]. Radical treatments of HCC by using surgical resection and/or liver transplantation may be curative but are suitable only for early-stage HCC [4]. Palliative therapy is commonly recommended for patients diagnosed with intermediate or advanced stages of HCC. Among the palliative therapies, multi-targeted tyrosine kinase inhibitors (TKIs), including sorafenib and lenvatinib, have been used for first-line systemic therapy after clinical trials demonstrated improved overall survival rates in patients with advanced HCC [5,6]. Although these agents have expanded the systemic treatment options for advanced HCC, a low tumor response rate and high toxicity warrants research towards development of new therapeutic alternatives. Chemotherapeutic agents, including doxorubicin, oxaliplatin, cisplatin, 5-fluorouracil, capecitabine, mitoxantrone, and gemcitabine, have been part of standard treatment regimens for the treatment of advanced HCC prior to the introduction of sorafenib [7]. However, they are no longer routinely used for patients in the advanced stages of the disease, due to adverse events and chemotherapy-related resistance caused by them [8].

The chemotherapeutic agent, gemcitabine is a nucleoside analog of deoxycytidine with a broad range of anti-tumor activities against various malignancies [9]. It has been widely studied as a single agent and in combination with a palliative therapy agent for the treatment of advanced HCC on account of its relatively lower toxicity compared to other chemotherapeutic agents [10,11,12]. However, the clinical benefit of gemcitabine is limited due to its short biological half-life, low bioavailability, rapid metabolic inactivation, and poor target specificity [13,14]. Various drug delivery systems, including nanoparticles, liposomes, aerosols, peptides, and aptamers, have been explored to overcome these drawbacks of gemcitabine [15,16,17,18,19]. Among the delivery systems, aptamer-drug conjugate (ApDC)-based targeted drug delivery has emerged as a potential strategy to improve the therapeutic efficacy and safety of chemotherapeutic agents [20]. Aptamers are chemically synthesized single-stranded oligonucleotides with a unique three-dimensional structure that can bind to a target protein with high specificity and affinity [21]. Moreover, aptamers can be directly combined with chemotherapeutic drugs via covalent conjugation or enzymatic incorporation [22,23]. Previous research in animals has shown that gemcitabine incorporated G-quadruplex aptamer has selective anti-proliferation activity in pancreatic cancer cells with low toxicity [24].

Glypican-3 (GPC3), a cell surface heparan sulfate proteoglycan, is an attractive therapeutic target and useful diagnostic biomarker because it is expressed at high levels in the plasma membrane of HCC cells, but not expressed in normal adult liver tissue [25,26]. Recent studies have shown that GPC3-targeted aptamers can specifically recognize the GPC3 protein expressed on the cytomembrane of HCC cells [27,28]. In an in vivo experiment, a fluorescent-labeled GPC3 aptamer displayed excellent potential as a tumor-targeting imaging probe. However, aptamer-drug conjugates targeting GPC3 for the treatment of HCC have not yet been reported.

In the present study, we developed a gemcitabine-incorporated GPC3-specific single-stranded DNA aptamer, G12msi, as an aptamer-based drug delivery vehicle for targeted therapy of advanced HCC. We evaluated the binding affinity, internalization property, and cytotoxicity of G12msi on GPC3-positive HCC cells. Furthermore, the therapeutic efficacy of G12msi in a murine xenograft HCC model was examined.

## 2. Materials and Methods 

### 2.1. Cell Lines and Reagents

The human hepatoma (HepG2, Hep3B and Huh7) and human epidermal carcinoma (A431) cell lines were purchased from American Type Culture Collection (Manassas, VA, USA). Cells were grown in Dulbecco’s modified Eagle’s medium (Life Technologies, Grand Island, NY, USA) supplemented with penicillin G (100 U/mL), streptomycin (100 μg/mL) and 10% fetal bovine serum. All cells were maintained at 37 °C in a CO_2_ incubator in a controlled humidified atmosphere composed of 5% CO_2_ and 95% air. Cell culture media, supplements, and serum products were purchased from Invitrogen (Carlsbad, CA, USA). All chemicals were used as received from Sigma-Aldrich (St. Louis, MO, USA).

### 2.2. Aptamers

The GPC3-specific aptamers were discovered using the SELEX process as described previously [29,30]. Briefly, a modified single-stranded DNA library with a 40-nucleotide random region (N40) containing either 5-[*N*-(1-naphthylmethyl)carboxamide]-2′-deoxyuridine (Nap-dU) or 5-(*N*-benzylcarboxamide)-2′-deoxyuridine (Bn-dU) in place of deoxythymidine was prepared. After each round of SELEX, binding assays were performed to measure the equilibrium dissociation constant (*K*_d_) values for each of the aptamers. To determine the minimal binding domain, full-length GPC3 aptamers were truncated to 35 nucleotides, named G12m, based on the secondary structure predicted by the Mfold and RNAstructure web server. Post-SELEX optimization was accomplished by incorporation of 2′-*O*-methoxy- and 2′-deoxy-2′-fluoro-modified nucleotides and 3′-inverted deoxythymidine (dT) into G12m aptamers. G12msi aptamer was generated by the incorporation of three gemcitabine phosphoramidite (5′-*O*-DMT-*N*^4^-benzoyl-2′,2′-difluoro-2′-deoxycytidine 3′-CE phosphoramidite) in the loop region of chemically modified aptamer G12ms. 

A random scrambled aptamer sequence was used as the non-specific binding control. Scrambled aptamer was designed to have the same number of nucleotides while displaying the different secondary structure as compared to G12msi aptamer. The sequence of the scrambled aptamer was 5′-AGAZZACZAGAZZCCGCGZCGZGZZGCACZZCAZA-idT (Z represents Bn-dU).

### 2.3. Serum Stability

G12m and G12msi were incubated with H4522 serum (Human AB serum, Sigma-Aldrich, St. Louis, MO, USA) at a final concentration of 50%. Samples were incubated at 37 °C, and aliquots were drawn at 0, 3, 6, 12, 24, 48, and 72 h after application. At each time point, a sample was taken and quick-frozen by placement in a deep freezer. Prior to analysis, aliquots were extracted using the phenol-chloroform-isoamyl alcohol (PCI) solution. Samples were analyzed by using denaturing polyacrylamide gel electrophoresis (PAGE) analysis. Aptamers were stained with SYBR™ Gold Nucleic Acid Gel Stain (Invitrogen, Carlsbad, CA, USA). The stained aptamer was imaged with a LI-COR Odyssey^®^ scanner and Image Studio^TM^ v5.2 software (LI-COR Biotechnology, Lincoln, NE, USA) and quantified using the Odyssey software as per the manufactur’s guidelines.

### 2.4. Flow Cytometry

Cells were grown at a concentration of 1 × 10^6^ m/L before the experiments were conducted. Cells were washed with Dulbecco’s phosphate-buffered saline (DPBS) and suspended in an ice-cold binding buffer. HepG2, Hep3B, Huh7, and A431 cells were incubated with 100 pM of fluorescently labeled G12msi or scrambled G12msi aptamers at 4 °C for 30 min. Cells were then washed with ice-cold binding buffer and flow cytometric analyses were performed on an LSR II flow cytometer (Becton Dickinson, Franklin Lakes, NJ, USA).

### 2.5. Confocal Fluorescence Microscopy

HepG2, Hep3B, Huh7, and A431 cells were grown in 3.5 cm glass-bottom dishes (MatTek Corporation, Ashland, MA, USA). To stain the plasma membrane, the cells were fixed in phosphate-buffered saline (PBS) containing 4% paraformaldehyde at room temperature. The fixed cells were incubated with 100 pM of fluorescently labeled G12msi or scrambled G12msi aptamers in a binding buffer DPBS supplemented with 4.5 g/L glucose, 5 mM MgCl_2_, 0.1 mg/mL yeast tRNA, and 1 mg/mL bovine serum albumin) at 4 °C for 30 min with gentle rocking. The cells were then rinsed with an ice-cold binding buffer to remove any unbound aptamer and counter-stained using Vectashield mounting medium with 4′,6-diamidino-2-phenylindole (DAPI) (1.5 μg/mL) (Vector Laboratories, Peterborough, UK). Confocal images were obtained using a Zeiss LSM-700 confocal laser scanning microscope (Carl Zeiss, Oberkochen, Germany).

For confocal z-stack imaging, HepG2, Hep3B, Huh7, and A431 cells were treated with 100 pM of Cy5-labeled G12msi and then incubated at 37 °C in 5% CO_2_ for 1 h. The cells were then rinsed with binding buffer and fixed in prewarmed 4% paraformaldehyde in PBS for 15 min at room temperature. The confocal z-stack imaging was carried out on a Zeiss LSM 770 and processed with ZEN 2010 image software (Carl Zeiss, Jena, Germany).

### 2.6. Lysosome Labeling

Lysosomes were labeled with Lysotracker green DND-26 (ThermoFisher Scientific, Waltham, MA, USA). Briefly, the cells were treated with 100 pM of Cy5-labeled G12msi and then incubated at 37 °C in 5% CO_2_ for 4 h. LysoTracker DND-26 (50 nM) was added to the cells 1 h before confocal imaging. After incubation, the cells were washed with PBS to remove any unbound aptamer and marker. Cells were fixed with 4% paraformaldehyde in PBS for 15 min at room temperature and counter-stained with DAPI (Vector Laboratories, Peterborough, UK). The cells were then visualized using a Zeiss LSM-700 confocal laser scanning microscope (Carl Zeiss, Oberkochen, Germany).

### 2.7. Cell Viability

Cytotoxicity of gemcitabine and G12msi was evaluated with a Cell Counting Kit-8 (CCK-8, Dojindo Laboratories, Kumamoto, Japan) according to the manufacturer’s specifications. The experiments were conducted with or without dipyridamole, a nucleoside transport inhibitor. Briefly, HepG2 and A431 cells were seeded into 96-well plates at a density of 1 × 10^4^ cells per well and incubated at 37 °C in 5% CO_2_. After 24 h, cells were incubated for 1 h at 37 °C in the serum-free medium which was then replaced with Krebs-Ringer-Phosphate-Hepes (KRPH) buffer containing the 10 μM dipyridamole. After 30 min of incubation, the cells were treated with different concentrations (0–1 μM) of G12msi or gemcitabine (Sigma-Aldrich, St. Louis, MO, USA) diluted in fresh cell culture medium for 4 h at 37 °C. The medium was replaced with fresh cell culture medium and then grown in a drug-free medium for 6 days. After 6 days, CCK-8 reagent was added to the culture medium of each well and incubated for 2 h. The absorbance of each sample was then measured at 450 nm to determine cell viability. The results were expressed as the mean percentage of cell viability relative to untreated cells. All experiments were performed in triplicate.

### 2.8. γ-H2AX Immunostaining

Double strand-break (DSB) induction and γ-H2AX immunostaining were performed using an OxiSelect™ DNA DSB Staining Kit (Cell Biolabs, Inc., San Diego, CA, USA) according to the manufacturer’s protocol. Briefly, the cells were seeded at 1 × 10^5^ cells per well in 8-well LAB-TEK^®^ II chamber slides™ (Nalge Nunc International, Hereford, UK). After 24 h, cells were treated with 500 nM of G12msi and gemcitabine in KRPH buffer and incubated for 4 h at 37 °C. The medium was replaced with fresh cell culture medium and cells were incubated overnight at 37 °C. After 24 h, cells were fixed with cold 3.7% formaldehyde in PBS and washed with cold PBS. The fixed cells were permeabilized in cold 90% methanol at 4 °C and blocked with 1% bovine serum albumin (BSA, ThermoFisher Scientific, Waltham, MA, USA) in PBS for 30 min at room temperature. The anti-phospho-histone H2A.X (Ser 139) antibody was added to the cells at a 1:100 dilution ratio in PBS containing 1% BSA for 1 hr. After washing with cold PBS, the cells were stained by fluorescein isothiocyanate (FITC)-conjugated secondary antibody at a 1:100 dilution ratio for 1 h at room temperature and then counter-stained with DAPI (Vector Laboratories, Peterborough, UK). The staining images of γ-H2AX were viewed using a Zeiss LSM-700 confocal laser scanning microscope (Carl Zeiss, Jena, Germany).

### 2.9. Cell-Cycle Analysis

HepG2 cells were treated with 500 nM of G12msi or gemcitabine diluted in KRPH buffer for 4 h at 37 °C. Following drug treatment, the medium was replaced with fresh cell culture medium and grown in a drug-free medium for 24 h. After 24 h, cells were stained with a mixture containing 50 μg/mL propidium iodide (PI) and 0.1 mg/mL RNase A for 30 min at room temperature. After incubation, the cell cycle distribution was analyzed using a BD FACSCalibur™ (Becton Dickinson, Franklin Lakes, NJ, USA). Data analysis was carried out with the CELLQuest™ (Becton Dickinson, Franklin Lakes, NJ, USA) and Modfit LT™ v2.0 software (Verity Software, Topsham, ME, USA) to quantify the fraction of viable cells in each phase of the cell cycle. All experiments were performed in triplicate.

### 2.10. Animal Model

Athymic female nude mice were purchased from Orient Bio Inc. (Gyeonggi-do, Korea). The animal experiments were performed with the approval of the Institutional Animal Care and Ethics Committee of Yonsei Laboratory Animal Research Center (approval No. 2018-0079, 15 November 2019). For the tumor model, a suspension of 1 × 10^6^ HepG2 cells in PBS was injected subcutaneously in the right thigh of 7-week-old Balb/c nude mice.

### 2.11. In Vivo Tumor Growth Inhibition

The in vivo therapeutic effect of G12msi was evaluated in mice bearing Capan-1 xenografts. When the tumor volumes reached approximately 200 mm^3^, HepG2 tumor-bearing mice were randomly separated into six groups (n = 5 per group) as follows: Group I, tumor control (DPBS); Group II, G12msi 50 mg/kg; Group III, G12msi 100 mg/kg; Group IV, G12msi 150 mg/kg; Group V, G12msi 200 mg/kg; Group VI, gemcitabine 18 mg/kg. The absolute amount of gemcitabine in 200 mg/kg G12msi was equivalent to 18 mg/kg gemcitabine. Gemcitabine and G12msi were dissolved in DPBS and then injected intravenously into mice eight times on alternate days. Body weight and tumor volume were measured on alternate days. Tumor volume was determined according to the following formula: tumor volume (mm^3^) = a × b^2^ × 0.52, where a was the longest diameter, and b was the shortest diameter. The mice were sacrificed on day 31 and the tumors were excised and weighed.

### 2.12. Statistical Analyses

Quantitative data were expressed as mean ± SD. Data were compared using Student’s t-test. *P*-values < 0.05 were considered statistically significant.

## 3. Results

### 3.1. Synthesis and Characterization of G12msi Aptamer

The GPC3-specific aptamers were identified using SELEX-based screening. Among the aptamer candidates, the G12 aptamer exhibited the highest affinity towards GPC3 protein, and the full-length 58 nucleotides G12 aptamer was truncated down to 35 nucleotides. The secondary structure predicted by Mfold web server indicated that the truncated G12m aptamer can form one hairpin stem-loop with one internal loop (Figure 1a). The binding affinity (*K*_d_ values) of G12m was 1.96 ± 0.36 nM (Figure 1b). The G12m aptamer was further modified with 2′-*O*-methoxy- and 2′-deoxy-2′-fluoro-modified nucleotides and 3′-inverted dT in the post-SELEX step. The chemically modified G12m aptamer, named G12ms, displayed a similar binding affinity (*K*_d_ = 1.33 ± 0.56 nM) with the G12m aptamer. The gemcitabine-incorporated G12msi aptamer was found to have a better affinity (*K*_d_ = 0.17 ± 0.13 nM) than its parent counterpart G12ms.

To investigate the nuclease resistance of aptamers, the stability of G12msi aptamer was evaluated in human serum. G12msi aptamer showed higher nuclease resistance than G12m aptamer (Figure 1c). G12msi aptamer has a half-life of 64 h, whereas the un-modified G12m aptamer has a half-life of 5.6 h in human serum at 37 °C.

### 3.2. Binding Specificity of G12msi Aptamer

The binding affinity of G12msi aptamer for GPC3-overexpressing HCC cells was confirmed by flow cytometric analysis using Cy5-labeled G12msi. The scrambled version of the G12msi aptamer was used as a negative control to determine non-specific binding. As shown in Figure 2a, the G12msi aptamer strongly bound to GPC3-positive HepG2, Hep3B, and Huh7 cells, whereas it bound weakly to GPC3-negative A431 cells. The results indicate that G12msi aptamer has a high affinity to the GPC3-overexpressing HCC cells.

The binding specificity of G12msi aptamer was further evaluated by confocal microscopy imaging. The strong fluorescence signal was observed mainly on the plasma membrane of GPC3-positive HCC cells, but not on GPC3-negative A431 cells (Figure 2b). Confocal microscopy showed that fluorescence signals from the scrambled G12msi aptamer were not detected in both GPC3-positive and GPC3-negative cells. These results demonstrate that the G12msi aptamer could selectively recognize the extracellular domain of GPC3 on the HCC cells.

### 3.3. Intracellular Localization of G12msi Aptamer

To investigate the internalization properties of G12msi, the GPC3-positive HCC cells and GPC3-negative cells were incubated with Cy5-labeled G12msi aptamer at 37 °C for 1 h. The fluorescence signal was mainly detected in the cytoplasm of HepG2, Hep3B, and Huh7 cells, but cytoplasmic fluorescence intensity decreased significantly in the A431 cells (Figure 3a). The scrambled G12msi aptamer was not readily detected in the cytoplasm of both GPC3-positive and GPC3-negative cells. Confocal z-stack images demonstrated that fluorescently labeled G12msi aptamers were efficiently internalized into GPC3-positive HCC cells; however, minimal internalization was observed in A431 cells.

The intracellular trafficking of G12msi aptamers was evaluated in HepG2 cells at different incubation time points using a LysoTracker™ Green. Binding of G12msi aptamers to HepG2 cells membrane was observed after 1 h of incubation (Figure 3b). G12msi aptamers gradually entered the HepG2 cells and subsequently extensive cytoplasmic accumulation of aptamers was observed after 4 h of incubation. Colocalization between late endosomes/lysosomes staining and G12msi aptamers was observed after a 1 h incubation and significant colocalization was detected after 4 h of incubation. The location and intensity of the fluorescence signal from the G12msi aptamer was similar to that from G12ms aptamer. These results indicated that most of the G12msi aptamers could enter the lysosomes of HepG2 cells.

### 3.4. Tumor Cell Cytotoxicity and DNA Damage Effects of G12msi Aptamer

To determine the effect of G12msi aptamer on cell proliferation, cell growth inhibition assays were performed on the GPC3-positive HepG2 cells and the GPC3-negative A431 cells. Cells were treated with increasing concentrations of G12msi aptamer or free gemcitabine. As shown in Figure 4a, G12msi aptamer significantly inhibited the cell proliferation in GPC3-overexpressing HepG2, with IC_50_ values of 179 nM. However, no significant cytotoxic effect of G12msi aptamer was observed in A431 cells. Free gemcitabine significantly inhibited both GPC3-positive and GPC3-negative cells.

To further investigate the induction of DNA double-strand breaks by G12msi aptamer, the level of phosphorylated H2AX (γ-H2AX) were evaluated by using immunofluorescence staining. The average number of γ-H2AX foci in HepG2 cells was significantly higher than that in A431 cells (*p* < 0.05) (Figure 4b).

### 3.5. Effects of Nucleoside Transporter Inhibition on G12msi Aptamer

The effect of nucleoside transporter inhibition on the anti-proliferative activity of G12msi aptamer was evaluated. HepG2 cells were treated with dipyridamole, a non-specific inhibitor of human equilibrative nucleoside transporter 1 (hENT1) transporters, prior to the addition of gemcitabine or G12msi aptamer. Treatment with gemcitabine inhibited the growth of HepG2 cells by 68% compared to the control group (Figure 5a). However, gemcitabine decreased the viability of HepG2 cells only by 13% in the presence of dipyridamole. The decrease in viability of HepG2 cells by dipyridamole was similar to that by gemcitabine in the presence of dipyridamole. For the HepG2 cells treated with G12msi aptamer, the number of viable cells decreased by 56% compared with the control group. In the presence of nucleoside transport inhibitor, the number of viable cells incubated with G12msi aptamer decreased by 34% compared with the control group. These results demonstrated that G12msi aptamer could generate GPC3-specific cytotoxicity in vitro.

### 3.6. Regulation of Cell Cycle by G12msi Aptamer

Treatment of HepG2 cells with 500 nM gemcitabine induced an increase in the percentage of cells in the G1 phase and a decrease in S phase and G2/M phase (Figure 5b). The percentage of cells arrested in the G1 phase increased from 42.7 to 65.3% after gemcitabine treatment. However, the treatment of gemcitabine with dipyridamole induced a decrease in the G1 phase population together with an increase in the S phase and G2/M phase. The cell cycle distribution in the HepG2 cells treated with gemcitabine and dipyridamole was similar to that in the control group. These data suggest that induction of cell death in HepG2 cells by gemcitabine is associated with the induction of G1 arrest.

An increase in the percentage of cells in the G1 phase together with a decrease in the S phase and G2/M phase was observed in G12msi treated HepG2 cells, while the percentage of cells that were arrested in G1 phase and S phase remained the same in G12msi and dipyridamole-treated cells compared to G12msi-treated cells. G12msi-treated HepG2 cells had 53.7% cells in the G1 phase compared to 53.3% in the G12msi and dipyridamole-treated cells. This data suggests that G12msi inhibited HepG2 cell proliferation after GPC3-specific internalization into HCC cells.

### 3.7. In Vivo Anti-Tumor Activity of G12msi Aptamer

To evaluate the anti-tumor efficacy of G12msi aptamer in vivo, HepG2 tumor-bearing mice were treated with DPBS, gemcitabine, and four doses of G12msi aptamer eight times on alternate days. As shown in Figure 6a, G12msi aptamer treatment dose-dependently inhibited tumor growth in the mice bearing GPC3-overexpressing HepG2 cells. Among all the G12msi-treated groups, the highest dose of G12msi aptamer generated the greatest anti-tumor effect. Administration of 200 mg/kg dose of G12msi aptamer resulted in approximately 56.5% inhibition in tumor growth on day 7 and 76.3% inhibition on day 15 as compared to control mice injected with DPBS. These differences were not statistically significant compared to the 18 mg/kg gemcitabine–treated groups. After stopping treatment at day 15, the average tumor size of the 18 mg/kg gemcitabine-treated group grew back from 185.6 ± 91.0 mm^3^ on day 15 to 330.1 ± 256.1 mm^3^ on day 31. On the contrary, the mean tumor size of the 200 mg/kg G12msi-treated group decreased from 201.3 ± 56.4 mm^3^ on day 15 to 123.2 ± 82.3 mm^3^ on day 31.

On Day 31, the average tumor weight was significantly reduced (*p* < 0.05) in the G12msi-treated group and the gemcitabine-treated groups as compared to vehicle-treated groups (Figure 6b). The mean tumor weight in the 200 mg/kg G12msi-treated group was 0.21 ± 0.09 g and the 18 mg/kg gemcitabine-treated group was 0.33 ± 0.05 g. The tumor weight of 200 mg/kg G12msi-treated group exhibited about a 3-fold decrease compared to that of 50 mg/kg G12msi-treated group (0.69 ± 0.12 g) and about a 4-fold decrease compared to that of the vehicle-treated groups (0.91 ± 0.28 g), respectively.

No significant difference in mean body weight was observed among the G12msi-treated groups (*p* > 0.05) (Figure 6c). However, administration of 18 mg/kg gemcitabine led to consistent decreases in body weight. On day 15, the body weight of 18 mg/kg gemcitabine-treated group decreased by 25% compared to the start day of treatment. After completion of treatment, the body weight of the 18 mg/kg gemcitabine-treated group was gradually regained and no significant difference in body weight was observed between groups after 10 days (*p* > 0.05).

## 4. Discussion

This study aimed to deliver gemcitabine specifically to GPC3-overexpressing HCC cells using an aptamer for the targeted therapy of advanced HCC. We designed and synthesized the GPC3-specific aptamer-drug conjugates, G12msi, by incorporation of gemcitabine into the GPC3 aptamer. In vitro experiments indicated that G12msi has a significantly high binding affinity to GPC3-positive HepG2, Hep3B, and Huh7 cells when compared to GPC3-negative A431 cells. G12msi can be selectively internalized into HepG2 cells overexpressing GPC3 and colocalized with late endosomes/lysosomes. An in vitro cytotoxicity study showed that G12msi significantly suppressed cell proliferation in GPC3-overexpressing HCC cells. In in vivo experiments, the G12msi-treated group exhibited significantly lower tumor volumes compared with the DPBS-treated group. G12msi suppressed the growth of HepG2 xenografts in a dose-dependent manner without toxicity.

Aptamers have many advantages over antibodies, including ease of generation, low batch-to-batch variation, versatile chemical modification, low immunogenicity and toxicity, and high physical and thermal stability [31]. However, in vivo application of aptamers is limited due to degradation by nucleases and rapid renal clearance [32]. Various types of chemically modified nucleotides have been developed to improve the physicochemical properties and efficacy of oligonucleotide therapeutics, including antisense oligonucleotides, small-interfering RNAs, aptamers, and microRNAs [33]. G12msi comprise chemically modified nucleotides, including Nap-dU, Bn-dU, 2′-*O*-methoxy nucleotides, 2′-deoxy-2′-fluoro nucleotides, and 3′-inverted dT. Nap-dU and Bn-dU are deoxyuridine-5′-triphosphate derivatives substituted at the 5-position of uracil nucleosides with naphthyl and benzyl groups, respectively. Inclusion of a hydrophobic group, including benzyl, naphthyl, tryptamino and isobutyl in uracil rings confers improved chemical diversity to aptamers and thus increases the chances of aptamer-target interaction [34]. Previous research has shown that incorporation of the Bn-dU into AS1411 aptamer significantly enhanced the targeting affinity to nucleolin expressing cancer cells [35]. Substitutions of the 2′-hydroxyl group on the sugar ring by 2′-amino, 2′-fluoro, or 2′-methoxy are the most common chemical modifications of nucleotides to improve nuclease resistance [36]. 2′-hydroxyl group modified RNAs are highly resistant to ribonuclease degradation because RNA hydrolysis proceeds by the nucleophilic attack of the deprotonated 2′-hydroxyl group on the adjacent phosphodiester bond of sugar-phosphate backbone [37]. Macugen^®^ is the first FDA-approved anti-vascular endothelial growth factor RNA aptamer containing 2′-*O*-methyl- and 2′-fluoro-modified nucleotides [38]. Modification at the 3′-end with biotin or inverted dT is also a widely used method to prevent degradation by 3′-exonucleases [39]. Aptamers modified with 3′-biotin or 3′-inverted thymidine showed significantly increased serum stability compared with the unmodified aptamers [40]. In this study, G12msi aptamer displayed higher nuclease resistance as compared to G12m aptamer. The high serum stability of G12msi can be explained by its additional 2′-*O*-sugar modifications and 3′-inverted dT capping. Incorporation of anti-cancer nucleoside analogs, such as gemcitabine and 5-fluorouracil in aptamer, is an attractive strategy for the development of an aptamer-drug conjugate. The current study shows that gemcitabine incorporation did not significantly affect the binding affinity to GPC3. Interestingly, the GPC3-binding affinity of G12msi improved after the incorporation of gemcitabine.

Gemcitabine has a short plasma half-life due to its rapid inactivation by cytidine deaminase expressed at high levels in blood and liver [41]. The rapid and extensive deamination into inactive metabolites may contribute to the low bioavailability of gemcitabine. We hypothesized that GPC3-targeted aptamers would specifically and safely deliver gemcitabine to GPC3-overexpressing HCC tumors, thus improving therapeutic efficacy. In the current study, we found that G12msi aptamer significantly reduces the growth of GPC3-positive HepG2 xenografts. The therapeutic efficacy of G12msi was comparable to that of gemcitabine administered at the same dosage. However, gemcitabine treatment caused significant weight loss in mice. The body weight of the G12msi-treated group, however, was not different from that of the untreated group (*p* > 0.05). A limitation of this study is that the cytotoxic effects of G12msi in chemoresistant cancer cells were not performed. Drug resistance has become a major hurdle in the chemotherapy of HCC [42]. In this study, nucleoside transporter inhibition assay showed that dipyridamole decreased the anti-proliferative activity of gemcitabine, but has little influence on the cytotoxic effect of G12msi against GPC3-positive HCC cells. Further studies are required to determine the therapeutic potential of G12msi against gemcitabine-resistant HCC cells.

## 5. Conclusions

GPC3 is abundantly expressed in HCC cells and serves as a potentially effective therapeutic target. In the present study, we selected the aptamer against GPC3 and three nucleotides were replaced with gemcitabine to generate the GPC3-specific aptamer-drug conjugate, G12msi. G12msi showed significant anti-tumor effects against GPC3-overexpressing HCC tumors both in the in vitro and in vivo models. The findings suggest that strategies to deliver chemotherapeutic drugs by using GPC3-specific aptamer might play a role in the treatment of HCC.

## Figures and Tables

**Figure 1 pharmaceutics-12-00985-f001:**
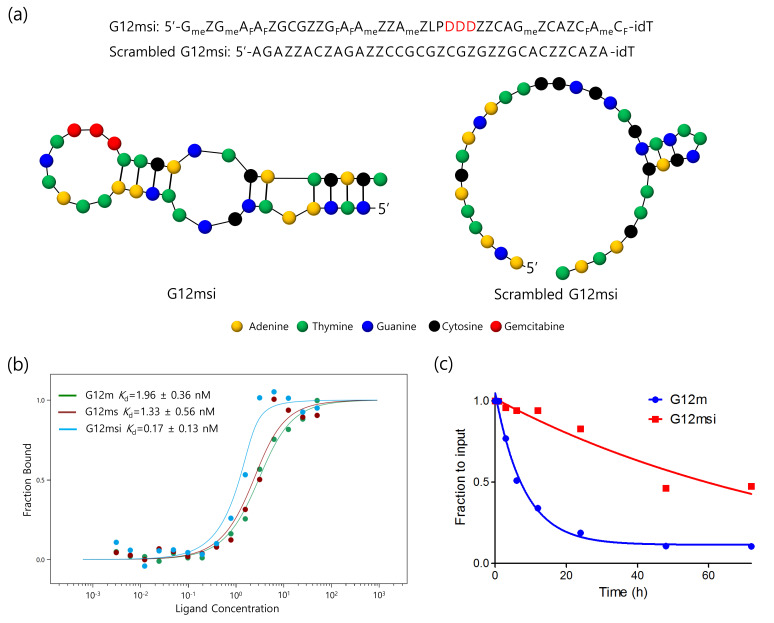
Characterization of G12msi aptamers. (**a**) The secondary structures of G12msi generated by RNAstructure (http://rna.urmc.rochester.edu/RNAstructure.html). The sequence of G12msi is shown with modifications indicated (P, 5-[*N*-(1-naphthylmethyl)carboxamide]-2′-deoxyuridine; Z, 5-(*N*-benzylcarboxamide)-2′-deoxyuridine; N_Me_, 2′-*O*-methoxy nucleotides; N_F_, 2′-deoxy-2′-fluoro nucleotides; L, C3-spacer; D, gemcitabine). (**b**) Determination of binding affinity of G12m, G12ms and G12msi aptamers to GPC3 protein. Equilibrium dissociation constants (*K*_d_) were determined by incubating GPC3 protein at varying concentrations of aptamer. (**c**) Nuclease resistance of G12m and G12msi aptamers in 50% serum. Aptamers were incubated with an equal volume of serum for 0, 3, 6, 12, 24, 48, and 72 h at 37 °C and analyzed by using polyacrylamide gel electrophoresis.

**Figure 2 pharmaceutics-12-00985-f002:**
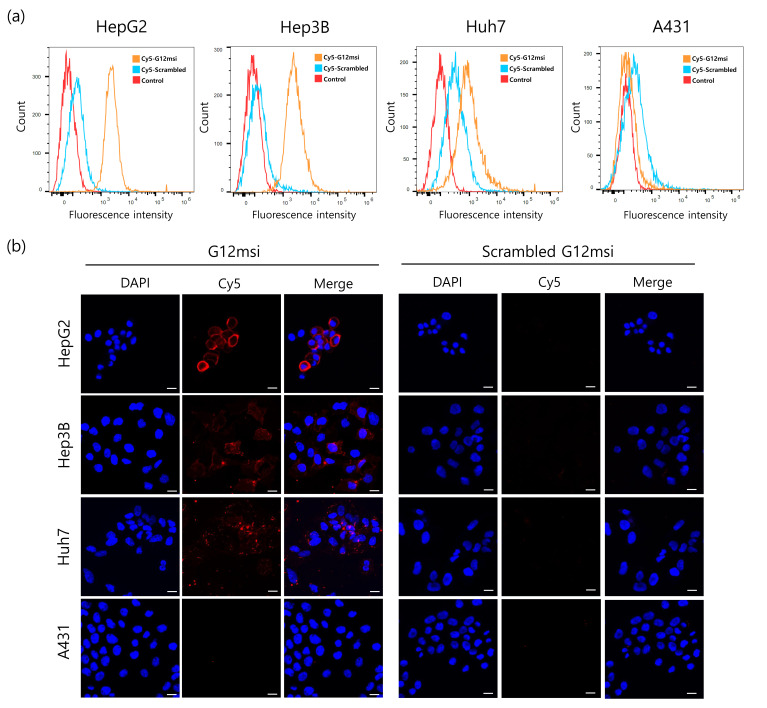
In vitro affinity and specificity of G12msi aptamers. (**a**) The specific binding capacity of G12msi to HCC cells was assessed by using flow cytometry. GPC3-positive HepG2, Hep3B, and Huh7 cell lines, and GPC3-negative A431 cells were stained with Cy5-labeled G12msi or scrambled G12msi. (**b**) Confocal microscopy analysis with Cy5-labeled G12msi or scrambled G12msi aptamers (Red) on GPC3 positive HepG2, Hep3B, Huh7 cells, and GPC3 negative A431 cells at 4 °C. The nuclei were stained with 4′,6-diamidino-2-phenylindole (DAPI, blue). Scale bars, 20 μm.

**Figure 3 pharmaceutics-12-00985-f003:**
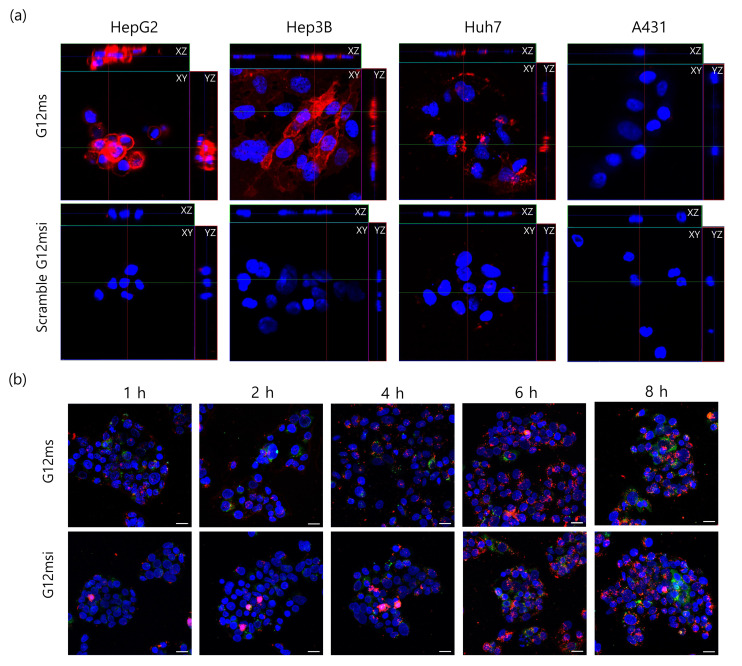
Internalization and intracellular localization of G12msi aptamers. (**a**) Representative z-stack images with orthogonal side-views of HepG2, Hep3B, Huh7, and A431 cells treated with Cy5-labeled G12msi or scrambled G12msi aptamers (red) at 37 °C. Nuclei were stained with DAPI (blue). (**b**) The cellular trafficking of G12ms and G12msi aptamers in HepG2 cells. Representative images showing the intracellular colocalization of the Cy5-labeled G12ms and G12msi aptamers (red) with a lysosomal marker (LysoTracker™ Green DND-26; green) by using confocal microscopy. The nuclei were counterstained with DAPI (blue). Scale bars, 20 μm.

**Figure 4 pharmaceutics-12-00985-f004:**
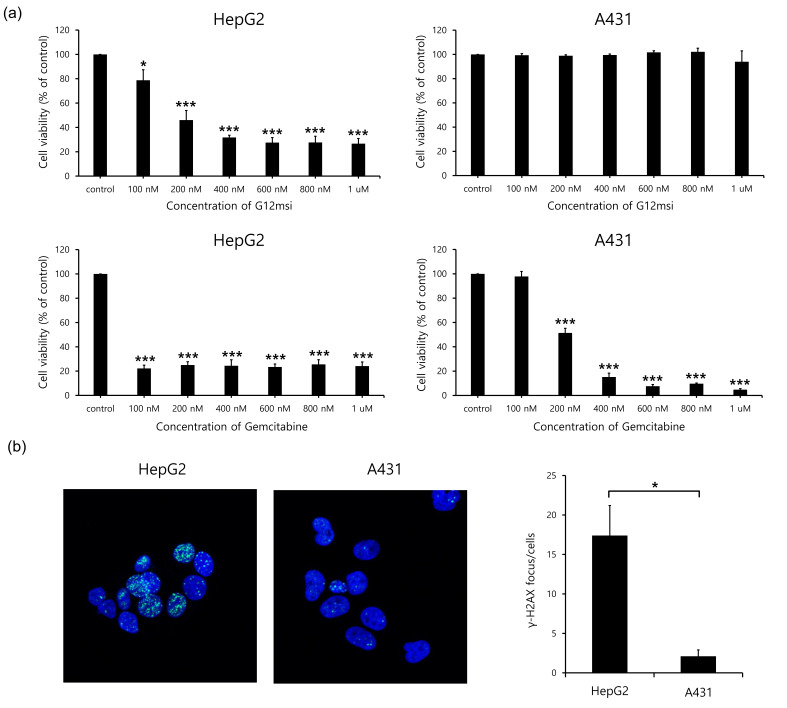
In vitro cytotoxicity of G12msi aptamer. (**a**) Cell viability analysis of GPC3-positive HepG2 and GPC3-negative A431 cells after treatment with increasing concentration of G12msi aptamer or gemcitabine. The relative amounts of viable cells were evaluated using the CCK-8 assay and cell viability (%) was normalized with untreated control cells. Data are shown as mean ± SD from three independent experiments. * *p* < 0.05, *** *p* < 0.0001, compared to control cells. (**b**) Representative confocal microscopic images of γ-H2AX nuclear foci (green) in response to G12msi aptamer. Nuclei were counterstained with DAPI (blue). Results represent the average of three independent experiments ± SD. * *p* < 0.05.

**Figure 5 pharmaceutics-12-00985-f005:**
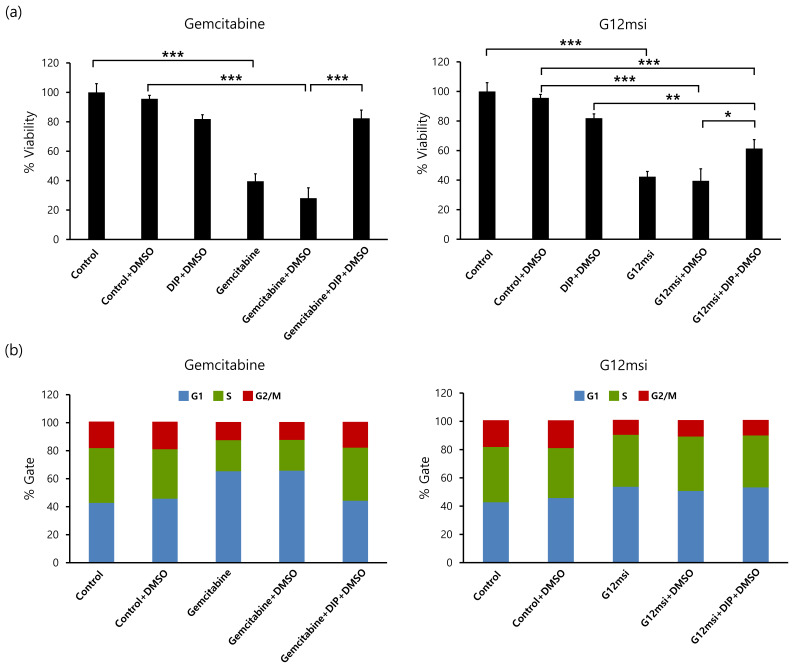
(**a**) Effect of nucleoside transporter inhibitor on the anti-proliferative activity of G12msi aptamer. Percentage of cell viability of G12msi aptamer and gemcitabine obtained after treatment of HepG2 cells with dipyridamole (DIP). Results are expressed as mean ± SD. * *p* < 0.05, ** *p* < 0.001, *** *p* < 0.0001. (**b**) Cell cycle analysis of HepG2 cells treated with G12msi and gemcitabine with or without dipyridamole. After staining with PI, cell cycle distribution was analyzed by using flow cytometry. (G1, gap1 phase; S, synthesis phase; G2/M, gap2/mitosis phase).

**Figure 6 pharmaceutics-12-00985-f006:**
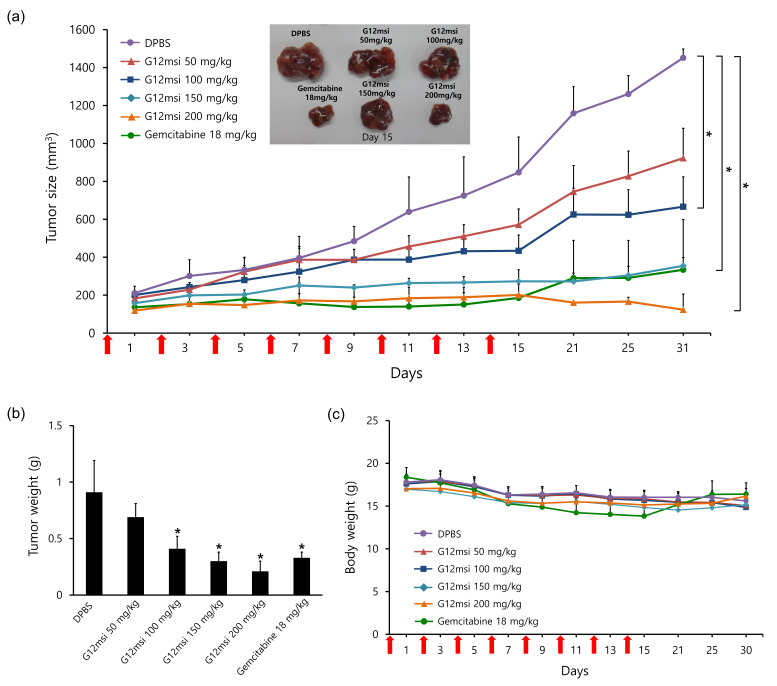
In vivo therapeutic efficacy of G12msi. (**a**) The growth curves of HepG2 tumor-bearing mice treated with intravenous injections of gemcitabine (18 mg/kg), G12msi (50 mg/kg, 100 mg/kg, 150 mg/kg, and 200 mg/kg), or vehicle (DPBS) alone. Representative image of tumors excised on day 15. * *p* < 0.05 (**b**) Changes in mean tumor weight and (**c**) the body weights of mice treated with gemcitabine, G12msi and vehicle. Red arrows indicate the time points of injection. Data are presented as the mean ± SD. * *p* < 0.05, compared to vehicle-treated group.

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
