# Peer review of "Targeted Therapy of Hepatocellular Carcinoma Using Gemcitabine-Incorporated GPC3 Aptamer"

_pharmaceutics, 2020, doi:10.3390/pharmaceutics12100985_

Round 1

Reviewer 1 Report

The obtained results described in this manuscript are impressive, I think the main conclusions are correct, but some important corrections are necessary to be implemented in the revised version. My major comments are as follows:

The colors in the schematic representation of used aptamer in Fig1 do not correspond with the legend. I do not believe that the secondary structure obtained by mfold corresponds with the real structure.

I did not find in the manuscript any notice, no details, concerning negative control… what is the “scrambled” aptamer. In my opinion, I recommend avoiding the term “scrambled”.

I recommend enlarging any legends, graph axes description, including inset of figures.

The legend in Fig 5b concerning details is missing to understand this figure.

Author Response

First, we would like to thank the reviewer for the detailed review of our manuscript. The manuscript has been revised according to the reviewers’ comments and suggestions. Below are our responses to the reviewers’ comments.

  1. The colors in the schematic representation of used aptamer in Fig1 do not correspond with the legend. I do not believe that the secondary structure obtained by mfold corresponds with the real structure.

Response: We apologize for this mistake. Guanine should have been a blue and adenine should have been a yellow. Thank you for your opinion. As per your suggestion, we have changed colors in the schematic representation in Figure 1 in the revised manuscript. In addition, we predicted secondary structure of G12msi aptamer using both Mfold and RNAstructure web server for the double confirmation. Figure 1a was acquired from RNAstructure web server. We have corrected the legend of Figure 1a and Materials and Methods section of the revised manuscript.

  1. I did not find in the manuscript any notice, no details, concerning negative control… what is the “scrambled” aptamer. In my opinion, I recommend avoiding the term “scrambled”.

Response: Thanks for this precious advice. To confirm the binding specificity of G12msi aptamer, we used scrambled aptamer as negative control. As per your suggestion, we have added the detailed information of scrambled GPC3 aptamer into the Materials and Methods section of the revised manuscript.

  1. I recommend enlarging any legends, graph axes description, including inset of figures.

Response: Thanks for the comment. Based on your comment, we have enlarged legends and graph axes description of figures in the revised manuscript.

  1. The legend in Fig 5b concerning details is missing to understand this figure.

Response: Thanks for this precious advice. As per your suggestion, we have added legend in Fig 5b. Thank you for your opinion.

Reviewer 2 Report

Manuscript reported “Targeted therapy of hepatocellular carcinoma using triple-gemcitabine-incorporated GPC3 aptamer”.

In this study, the GPC3-targeted aptamers were selected by the SELEX process, and the chemotherapy drug gemcitabine was internally incorporated into the aptamer sequence. In vitro studies demonstrate that synthetic gemcitabine-aptamers selectively bound GPC3-positive tumor cells and induced cell death. In addition, animal model study showed that systemic administration of gemcitabine-aptamers inhibited growth of xenograft tumors derived from GPC3-positive hepatocellular carcinoma cells. Findings suggest that synthetic gemcitabine-aptamers may be a promising strategy for targeted cancer therapy.

Study is well designed and results are solid.

Manuscript may be further improved by address following minor issues.

  1. It will be interesting to see primary aptamer sequence developed by SELEX, rationale and design of gemcitabine incorporation, and structural comparison of them. Such information will be useful for other investigators to design new aptamer-drug conjugates. Date can be submitted as supplemental figures.
  2. There is no point to add the word of “triple” in title since no relevant study were described.
  3. In addition to Kd with GPC3 proteins, what are Kd of aptamers with GPC3-positive tumor cells?
  4. Please add labels of X and Y axis of figure 2a
  5. Figure 4a needs statistical results/labels, similar to that showed in Figure 4b.
  6. Please revised Figure 5a similar to that showed in Figure 4b.
  7. Please use color graph in Figure 5b and label individual portions of cycle.
  8. Please add statistical labels in Figure 6.
  9. Please indicate time points of treatments in Figure 6
  10. Please add statistical results/labels similar to that showed in Figure 4b

Author Response

First, we would like to thank the reviewer for the detailed review of our manuscript. The manuscript has been revised according to the reviewers’ comments and suggestions. Below are our responses to the reviewers’ comments.

  1. It will be interesting to see primary aptamer sequence developed by SELEX, rationale and design of gemcitabine incorporation, and structural comparison of them. Such information will be useful for other investigators to design new aptamer-drug conjugates. Date can be submitted as supplemental figures.

Response:  We have synthesized GPC3-specific aptamers using the SELEX process as described previously. For the incorporation of gemcitabine into the aptamer, we used gemcitabine phosphoramidite (5’-O-DMT-N4-benzoyl-2’,2’-difluoro-2’-deoxycytidine 3’-CE phosphoramidite). In accordance with the reviewers’ comments, we have added the more information into the Materials and Methods section of the revised manuscript. However, we had to briefly describe the synthesis method of G12msi aptamer due to the patent held by Aptamer Science Inc.

  1. There is no point to add the word of “triple” in title since no relevant study were described.

Response: We agree with your opinion. In accordance with the reviewers’ comments, we have removed the word of “triple” in title and abstract in the revised manuscript. We thank the reviewer.

  1. In addition to Kd with GPC3 proteins, what are Kd of aptamers with GPC3-positive tumor cells?

Response: Thank you for the comment. Unfortunately, we didn’t measure the Kd values of aptamers with tumor cells. However, the specific binding of G12msi was verified by confocal fluorescence microscopy and flow cytometry using fluorescently labeled G12msi or scrambled G12msi aptamers. G12msi aptamer strongly bound to GPC3-positive cells, whereas it bound weakly to GPC3-negative cells. In addition, scrambled G12msi aptamer were not detected in both GPC3-positive and GPC3-negative cells in this study.

  1. Please add labels of X and Y axis of figure 2a

Response: Thanks for this precious advice. We have added labels in flow cytometry graphs of figure 2a in the revised version. We thank the reviewer.

  1. Figure 4a needs statistical results/labels, similar to that showed in Figure 4b.

Response: Thank you for your opinion. Based on your comment, we have added the statistical results/labels in Figure 4b of the revised manuscript.

  1. Please revised Figure 5a similar to that showed in Figure 4b.

Response: Thank you for your opinion. Based on your comment, we have added the statistical results/labels in Figure 5a of the revised manuscript.

  1. Please use color graph in Figure 5b and label individual portions of cycle.

Response: Thank you for the comment. We agree to this comment. In accordance with the reviewers’ comments, grayscale graph of Figure 5b was changed to color graph in the revised version. We thank the reviewer for the suggestion.

  1. Please add statistical labels in Figure 6.

Response: Thank you for your opinion. Based on your comment, we have added the statistical results/labels in Figure 6a in the revised version.

  1. Please indicate time points of treatments in Figure 6.

Response: We agree to this comment. In accordance with the reviewers’ comments, we have added the time points of treatments in Figure 6a of the revised manuscript.

  1. Please add statistical results/labels similar to that showed in Figure 4b

Response: We agree to this comment. Based on your comment, we have added the statistical results/labels in Figure 6b in the revised version. We thank the reviewer for the suggestion.

Reviewer 3 Report

In this manuscript, the authors developed a triple-gemcitabine-incorporated GPC3-specific single stranded DNA aptamer, G12msi, for targeted therapy of advanced HCC tumors. The anticancer effects were tested in vitro and in vivo using preclinical models. A specific effect against GPC3-positive cells was observed in vitro. In my opinion, data support the authors' conclusions.

I guess if the authors performed a systemic investigation of the side effects of G12msi in vivo.

Author Response

First, we would like to thank the reviewer for the detailed review of our manuscript. The manuscript has been revised according to the reviewers’ comments and suggestions. Below are our responses to the reviewers’ comments.

I guess if the authors performed a systemic investigation of the side effects of G12msi in vivo.

Response: Unfortunately, the systemic investigation of toxic effects such as hematological and biochemical analyses were not performed in this study. We monitored the risk of toxicity by measuring the body weight of mice. In this study, the body weight data of 18 mg/kg gemcitabine-treated group clearly showed the toxicity of gemcitbine. The body weight of 18 mg/kg gemcitabine-treated group decreased by 25% compared to the start day of treatment. However, no significant difference in mean body weight was observed among the G12msi-treated groups. Thank you for your opinion.

Round 2

Reviewer 1 Report

Authors  have implemented changes which help understanding of the content. I reccomend to accept revised form.